# Spatiotemporal Characteristics of the Mud Receiving Area Were Retrieved by InSAR and Interpolation

**Bo Hu \* and Zhongya Qiao**

Surveying Engineering, School of Civil and Transportation Engineering, Guangdong University of Technology, Guangzhou 510006, China
\* Correspondence: hubo@gdut.edu.cn

**Abstract:** The mud receiving area is an important sand storage area for dredging sea sand reclamation and sand-dumping in the waterway. The sediment accumulation area generated in the process of sand dumping and sand storage has an impact on the surrounding transportation facilities and the normal use of the entire sand storage area. From 6 August 2021 to 9 May 2022, The Sentinel-1A 24-view SLC data covering the sludge area were used to monitor the safety around the seawall road by InSAR technology. Synthetic aperture radar differential interferometry (Differential InSAR, D-InSAR) technology can obtain surface micro deformation information through single-time differential interference processing, mainly used for sudden surface deformation. D-InSAR technology detected five accumulation areas with a thickness of more than 10 cm near the seawall road, earth embankment, and cofferdam, and TS-InSAR (Time series InSAR) technology was used to retrieve the deformation of the surrounding road. The road settlement is a slight settlement distributed between $\pm 5$ mm/a. This paper uses the leveling results combined with variance analysis to verify the fusion of different TS-InSAR methods while considering the area of data loss due to causes such as loss of coherence. This paper also considers the common ground continuity and uses the adjacent interpolation and bilinear interpolation algorithm to improve knowledge of the study area seawall road and the surrounding soil embankment deformation data of the road. Compared with the leveling data, the difference between the missing data and the leveling data after interpolation is stable at about 1–7 mm, which increases the risk level of part of the road which needs to be maintained. It provides a reference method to make up for the missing data caused by ground incoherence.

**Keywords:** D-InSAR; TS-InSAR; fusion; accumulation; sedimentation; ANOVA; nearest; bilinear

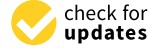



## 1. Introduction

Adi, N. and Albert, P. et al. [1–3] have shown that the dredging of seabed sediment has become an increasingly important work for port maintenance. In large dredging projects such as channel dredging and sea reclamation, sand storage and sediment accumulation areas are often set up. Sand-throwing vessels often take sand from the open sea and throw it into the sand storage area, and then the cutting-suction dredgers blow it onto the shore. In the combined construction of throwing and blowing, the core process is to set up a sand storage area in the sea near the filling area and complete the construction process of throwing and blowing sand by means of a sand ship and a cutter suction dredger. To ensure that the ship and waterway can be used normally, the shoal area needs to be dredged. The mud area is an important reservoir in the dredging sand area. Its uneven sediment accumulation is caused by sand reclamation and the extrusion pressure on its surrounding environment. The area of land reclamation works normally, and the mud area needs perimeter security. The impact of land reclamation on temporal and spatial dynamic monitoring needs to be better understood.

GRACE provides users with gravity products, which can be used to detect the changes in the earth's surface mass caused by its surface sediments [4]. GRACE has certain limitations in solving the problem of sediment deposition. Due to its low spatial resolution

(about 250 KM), it cannot meet the requirements of precision measurement of sediment deposition in small areas. Traditional machine learning methods can calculate and estimate sediment deposition volumes using neural networks and regression analysis [5], which requires a large amount of time to train samples, and it is difficult to directly and accurately express the spatial and temporal distribution of the accumulation area.

Surface deformation monitoring methods based on surveying cannot elucidate the internal layout of the land in the mud area. Interferometric Synthetic Aperture Radar (InSAR) technology is advantageous because it can be used at any time and can efficiently illuminate the surface's spatiotemporal characteristics. In 2014, the European Space Agency (ESA) launched the spaceborne SAR satellite Sentinel-1A/B and provided the TOPS topographic observation data covering the whole world free of charge to users. Compared with TerraSAR-X, COSMO-SkyMed, the acquisition cost is low, the method is simple, the influence degree of chimerization is high, and the accuracy can meet the public's demands. At present, InSAR technology has been widely used in the settlement monitoring of large areas [6–8] and earthquake-related research. InSAR is used to retrieve the motion law of the ground slip zone [9,10]. Long-term over-mining leads to obvious subsidence and cracks in the ground, threatening the safety of buildings and surrounding facilities. InSAR can be used to do a good job in the early warning of settlement areas [11].

D-InSAR technology [12–14] can be used to detect the sediment accumulation area and diffusion characteristics of the mud receiving area quickly. The permanent scatterer interference superposition technology (PS-InSAR) can obtain the deformation of different scatterers based on the echo signal intensity of ground scatterers [15,16]. Small Baseline set interferometric superposition technology (SBAS-InSAR) uses SAR images of different phases to obtain the filter phase (SDFP) based on coherence through cross processing and then expresses it in the form of pixels. Each pixel records the deformation information of the location [17–21]. The two interference superposition techniques have their own advantages. At present, some scholars classify the permanent scatterer points with high coherence by using the clustering analysis algorithm. SBAS-InSAR vector data is used in the sparse area of PS points to obtain the relatively optimized land surface settlement [22]. The integration of various InSAR technologies [23,24] and optimization has become the mainstream development direction of InSAR.

It is difficult to solve the atmosphere errors, ionosphere errors, orbit errors, and coherence errors that largely affect the integrity and precision of the measurement results. This paper puts forward a kind of interpolation method based on the pavement continuity of the ground settlement data of quadratic optimization, excluding the existence of absolute faults. The ground can be regarded as a continuous curved surface function [25,26]. The fused InSAR vertical ground deformation after the first optimization is used to optimize the deformation data by the second interpolation to improve the integrity of the data. A model matrix construction based on polynomial interpolation usually includes the nearest neighbor in conjunction with bilinear and bicuxic algorithms, among which the nearest neighbor interpolation algorithm can provide a simpler and more realistic image reconstruction for sparse data [27] and has been well supported in the evaluation [28], while the bilinear interpolation generally meets the interpolation of missing pixels [29]. On the premise of satisfying the SAR satellite resolution, different interpolation methods are well applied to fused MT-InSAR. In the whole interpolation process, the GIS-based buffer analysis [30] provides the basic data points for interpolation, and the region of interest is obtained through buffer analysis. The raster vector data of the buffer is generated according to the size of the corresponding radar resolution pixel. The raster turn point can be used as the interpolation base point. The vector pixel with settlement data obtained by SBAS-InSAR and the permanent scatterer PS point obtained by PS-InsAR are used as the interpolation reference point.

Analysis of variance (ANOVA) is also known as 'F' test, variance analysis can be used for more than two, and two samples of difference significance test, the current has been applied to transformer fault in [31], is also used to analyze the effect of food composition

between [32], a growing number of studies show that the method in the interdisciplinary generality, within the scope of analysis of variance can also be very good solve the problem of two kinds of InSAR technology matching fusion, Data loss caused by ground incoherence can be remedied by fusion interpolation.

The leveling method and accuracy have been continuously improved, and there are more opportunities to expand in the field of remote sensing. In this paper, data from 30 leveling monitoring points in the study area were used to verify the interpolation accuracy of road deformation data. In the traditional InSAR technology, the data volume is increased while the accuracy is guaranteed.

In this paper, D-InSAR and time-series InSAR technology are used to invert the spatiotemporal characteristics of the mud receiving area, monitor the changes of the sand storage area and the sedimentation of the surrounding roads during the dredging process of the waterway, provide a reference method for the analysis of accumulation phenomenon, and fuse the settlement data by ANOVA, and impolate the data loss caused by phase incoherence, which is expected to be applied to the relevant areas of ship waterways dredging and poor coherence.

## 2. Materials and Methods

### 2.1. Study Area

The hinterland of Jiangmen City, where the XinZhouwei mud receiving area project is located, is vast, located at 22°07′ N–22°09′ N, 113°02′ E–113°05′ E, this is shown in Figure 1. It is the gateway of western Guangdong Province and an important node city in the Guangdong-Hong Kong-Macao Greater Bay Area. After the completion of the channel project where the mud area is located, the section of the Huangmao sea operation Area will meet the requirements of full tide two-way navigation of 10,000-ton ships.

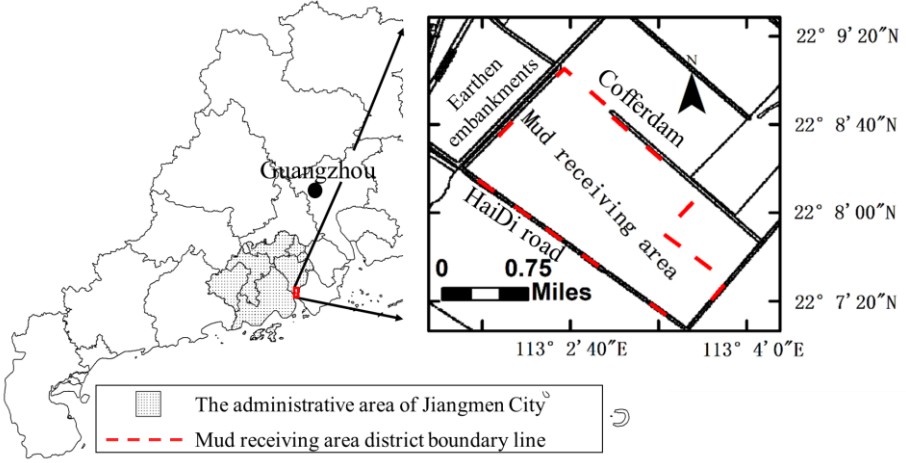

**Figure 1.** The geographical location of the study area.

The SAR data were obtained from 24-view C-band Sentinel-1A data of the European Space Agency covering the Mud receiving area of Xinzhouwei area in Jiangmen City. The flight track was orbit ascending data. The imaging mode was IW interference wide band (TOPS) mode, and the polarization mode was co-polarized VV, with an incident Angle of 20.84°. The range resolution is 2.32956 m, the azimuth resolution is 13.9544 m, and the ground range resolution is 3.6993 m. Japan Aerospace Exploration Agency (JAXA) provides a high-precision global digital surface model using high-resolution reference DEM [33]. The horizontal resolution of the data is 30 m.

In order to detect the implementation of the sea sand blowing and filling project and the settlement of the HaiDi road in the Xinzhouwei mud receiving area in real time, the interference superposition timing analysis method was used to obtain the settlement rate for the settlement safety assessment of the surrounding seawall road, that is, Time series

In-SAR (TS-InSAR) technology [34,35]. The monitoring and control standards of this region are designed according to the Standards for Geological Disaster Risk Assessment, Highway Subgrade Design and Planning, and Indicators System and Technical Methods for Evaluation, Monitoring and Early Warning of Geological Environment Bearing Capacity [36,37], as shown in Table 1.

**Table 1.** Settlement monitoring control values and Ground settlement rate grading standard.

| Project | The Control Value of the Vertical Displacement | | | | |
|---|---|---|---|---|---|
| | Absolute Value (mm) | Rate of Change (mm/Month) | Cumulative Value/mm | | |
| Earthen embankment | 10 | ±10 | ±10 | | |
| | 10 | ±10 | ±10 | | |
| HaiDi road | 10 | ±10 | ±10 | | |
| **Level** | **I** | **II** | **III** | **IV** | **V** |
| Sedimentation rate (mm/a) | 0–10 | 10–15 | 15–20 | 20–25 | >25 |
| Degree grading | slight | average | severe | More severe | Extremely serious |

Since 10 August 2021, the auxiliary settlement monitoring has been carried out in the XinZhouWei mud receiving area. The vertical displacement monitoring reference network is arranged as a closed leveling route, and the HaiDi road settlement monitoring in mud absorbing area has been carried out according to the technical requirements of the "China National Standards for Levelling Measurement of 3rd and 4th Class" GB/T 12898-2009. The error in elevation is ±2 mm. In order to prevent the level monitoring points from being affected by external environmental factors and ensure high backscattering intensity, the level monitoring points are placed at the side of the concrete seawall road for precision monitoring. The location and number of level reference points are shown in Figure 2.

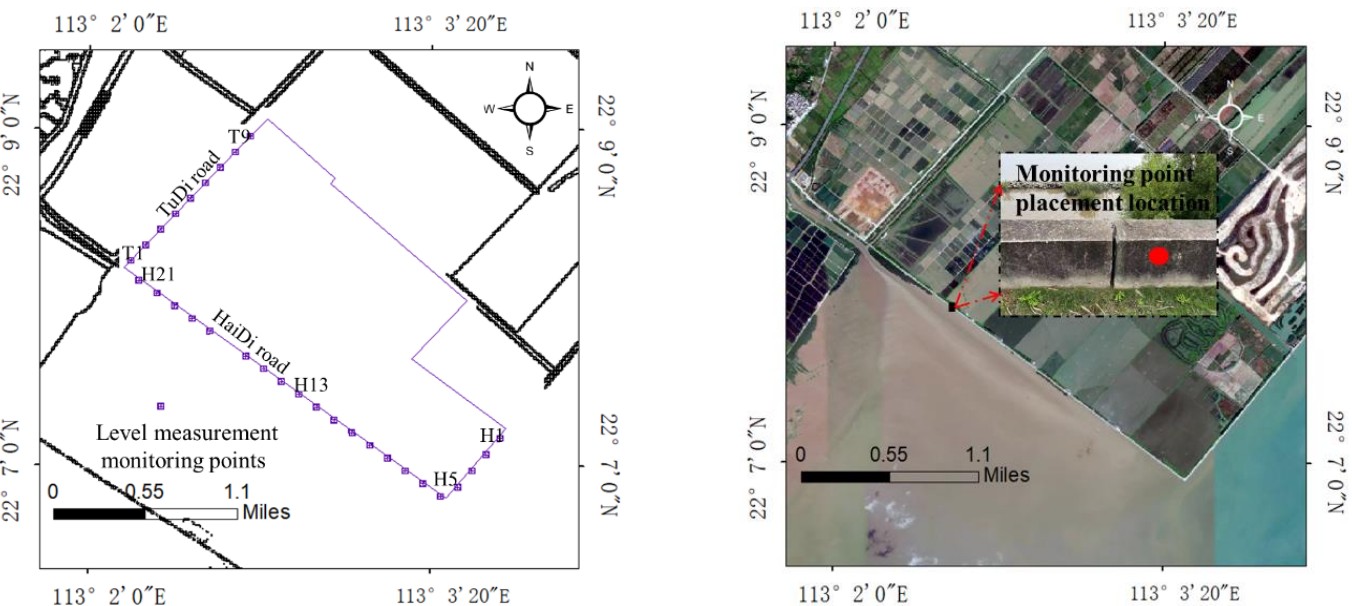

**Figure 2.** 21 level monitoring points along the road (The red dot is where the horizontal base point is laid).

*2.2. Analysis Method*

2.2.1. Differential Radar Interferometry (D-InSAR) and TS-InSAR

Differential radar interferometry technique (D-InSAR) can use not the same area at the same time the differential interference in the SAR image data, the small surface deformation can be detected. The experiment adopts the 2-pass difference method for processing:

The distance direction of the radar to the target, also known as the satellite line-of-sight direction (LOS), the radar can obtain the slant range by detecting the echo signal in this direction, and the deformation information in the oblique direction is projected to the earth's surface, which can be converted into the real deformation information on the ground vertical direction (Ground range). If the deformation in the direction of SAR observation skew distance (Slant range) in the two SAR image information is set as $\delta r$, the interference phase $\varphi_1$ can be expressed as:

$$\varphi_1 = -\frac{4\pi}{\lambda}[(r_1 + \delta r) - r_2] \tag{1}$$

$$\varphi_0 = -\frac{4\pi}{\lambda}[r_1 - r_2] \tag{2}$$

$$\varphi_1 - \varphi_0 = -\frac{4\pi}{\lambda}\delta r \tag{3}$$

The phase $\varphi_0$ is derived from the phase obtained by DEM simulation using interference baseline and incident Angle. In the actual differential processing process, interference phase still needs to remove the ground level effect, and the simulated phase only needs to simulate terrain phase. This method can detect and analyze large area and high deformation area with only 2 spoke SAR images.

Multi-temporal interference superposition (MT-InSAR) technology can obtain milli meter-level information of the surface by mining regional pixels within the range of SAR images that can recognize time series [38], and using the performance law of ground deformation in phase within a certain time. Among them, permanent scatterer differential interferometry (PS-InSAR) can effectively overcome the errors caused by D-InSAR spatial and temporal decoherence and atmospheric delay, significantly improve the monitoring accuracy of ground deformation, and play an important role in monitoring the protection system around the sludge area. The seawall road (HaiDi road) guardrail is a concrete structure with strong backscattering intensity. Its stability can be processed by PS-InSAR, SBAS can significantly increase the number of effective interference image pairs. The combination of the two can optimize the processing of interference image pairs, and the purpose of encrypting deformation data can be realized by the fusion of two interference superposition technologies. Some areas of ground incoherence can also have change rules. The demand can be improved by optimizing the InSAR data processing results. Its principle and processing flow are as shown in Figure 3.

PS-InSAR technology selects the master image by comprehensively evaluating the spatio-temporal baseline of multi-temporal radar images in the same area, and the rest are slave images. The master and slave images are registered and the interferogram is generated. The interference phase $\varphi_i$ is:

$$\varphi_i = \varphi_{top0} + \varphi_{flat} + \varphi_{orbit} + \varphi_{defo} + \varphi_{atmo} + \varphi_{scat} + \varphi_{noise} \tag{4}$$

where, $\varphi_{top0}$ is the terrain phase; $\varphi_{flat}$ is flat phase; $\varphi_{orbit}$ is the orbit error phase; $\varphi_{defo}$ is the deformation phase; $\varphi_{atmo}$ is the atmospheric delay phase; $\varphi_{scat}$ is the phase generated by the change of scattering characteristics of the ground point target. $\varphi_{noise}$ is the noise phase, and the phase difference of adjacent target points in the differential interferogram can be expressed as follows.

$$\Delta\varphi = \frac{4\pi \cdot t \cdot \Delta v}{\lambda} + \frac{4\pi B_\perp \Delta\varepsilon}{\lambda R sin\theta} + \Delta\omega + 2n\pi \tag{5}$$

where $\Delta v$ is the difference of linear deformation rate; $\Delta \varepsilon$ is elevation error increment; $\Delta \omega$ is the sum of nonlinear deformation phase, atmospheric delay and noise phase. According to the generated M-amplitude interferogram, a coherence coefficient can be set:

$$\gamma = \frac{1}{M} \sum_{i=1}^{M} exp\left\{ j\left[ \Delta\varphi - \left( \frac{4\pi \cdot t \cdot \Delta v}{\lambda} + \frac{4\pi B_\perp \cdot \Delta\varepsilon}{\lambda R sin\theta} \right) \right] \right\} \qquad (6)$$

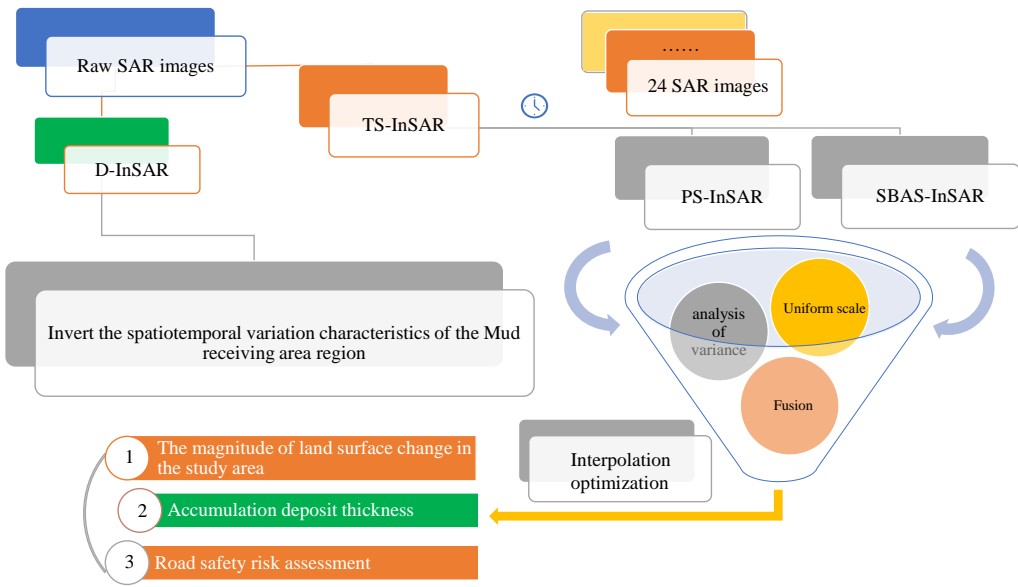

**Figure 3.** D-InSAR and TS-InSAR processing flow and product optimization processing method.

According to the distribution characteristics of noise phase, atmospheric delay phase and nonlinear deformation phase in space and time, the nonlinear deformation phase and random noise phase are filtered by high-pass filtering in time domain and low-pass filtering in space domain respectively. The filtered phase is added to the atmospheric phase of the relative main image. The atmospheric delay phase of PS point on each image is calculated as follows:

$$\left\{ \begin{array}{c} \varphi_{atm} = \left[ (\omega_i)_{HP_{time}} \right]_{LP_{space}} + (\overline{\omega_i})_{LP\_space} \\ \varphi_{nonlinear} = (\omega_i - \varphi_{atm})_{LP\_time} \end{array} \right\} \qquad (7)$$

Then, the nonlinear deformation phase can be obtained by low-pass filtering the residual phase of the delayed phase of the atmosphere in the time domain, and then the deformation phase can be obtained. Finally, the surface deformation rate can be solved. SBAS forms a short baseline SAR image set by connecting the SAR images that are independent of each other caused by the long baseline to increase the sampling rate of data acquisition, so as to form several small sets in the existing SAR image data set.

2.2.2. Analysis of Variance for Randomized Block Design

In order to verify the feasibility and accuracy analysis of the two interference superposition monitoring methods, PS-InSAR, SBAS-InSAR and the common part of leveling results were analyzed by ANOVA, and the variance of randomized block design was used for analysis. For a given significance level $\alpha$, F corresponds to a critical value of $F_\alpha$ when the following conditions are met:

$$F = \frac{MSTR}{(MSE)} > F_\alpha \qquad (8)$$

We can reject the null hypothesis to test the differences between different methods, Randomized block design, also known as compatible block design, is an extension of

paired design. The specific method is: first, according to the monitoring point number and measurement method of non-treatment factors that affect the experimental results, the experimental subjects are divided into blocks, and then the experimental subjects in each block group are randomly assigned to each treatment group or control group.

Compared with the completely randomized design, the randomized block design has the characteristic that the number of random assignments should be repeated several times, and each random assignment is carried out on the experimental subjects in the same block, and the number of experimental subjects in each treatment group is the same, which is balanced within the block. In the statistical analysis, the sum of squared deviations of block variation was separated from the sum of squared deviations within a completely randomized design, so as to reduce the sum of squared deviations within a group (sum of squared errors) and improve the efficiency of statistical testing.

### 2.2.3. Interpolation and Optimization

Due to ground subsidence and uplift for continuous, the height that is continuous change (not including absolute fault) continuous curved surface, the surface in office a little in latitude, longitude unchanged, or decreases with increase of latitude (longitude) is continuous, the settlement of the ground (the road) to its height or lift the quantity also in continuous change, as shown in Figure 4I, The projection curve equations of the three surfaces obtained from the ground surface equation $Z = F(X, Y)$ are $Z = F(X)$, $Z = F(Y)$ and $Y = F(X)$ respectively. Therefore, the surface condition and change rule can be determined by reasonable interpolation according to the numerical law of the grid points.

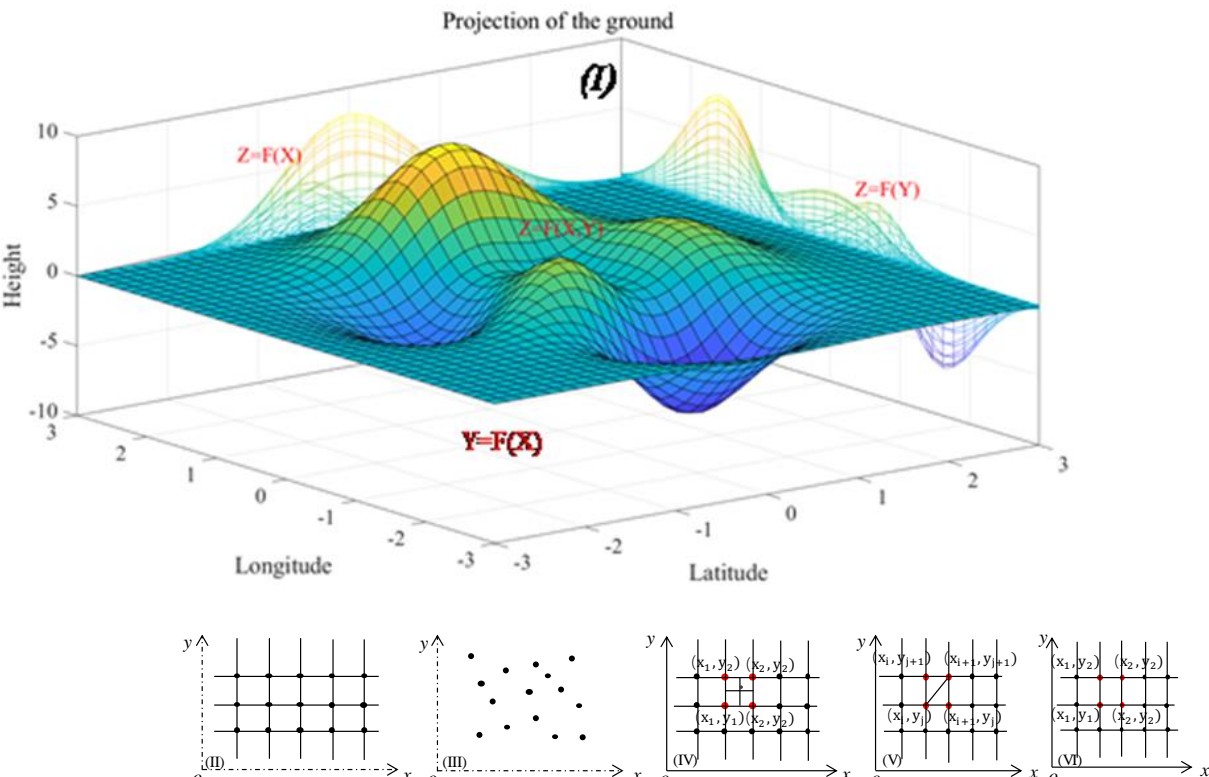

**Figure 4.** The continuous curve of the continuous ground surface projected on the XOZ and YOZ surfaces (**I–III**) are the regular node and the scattered node of the interpolation node, respectively. (**IV–VI**) are the nearest neighbor interpolation, process of piecewise linear interpolation and bilinear interpolation.

For this kind of problems, high-dimensional interpolation method is often used, and the data is assumed to be correct. It is required to describe the situation between the data

points in a certain way, so as to solve the problems encountered in actual production and life. Among them, the most commonly used high dimensional interpolation is two-dimensional interpolation. The basic interpolation principle is as follows: In this paper, the raster data obtained by InSAR processing is extracted from its raster center, and the dataset

$$\{(x_1, y_1, z_1), (x_2, y_2, z_2), \ldots, (x_n, y_n, z_n)\} \tag{9}$$

it is using known in point set $\{(x_1, y_1, z_1), (x_2, y_2, z_2), \ldots, (x_n, y_n, z_n)\}$. The function value $\{z_1, z_2, \ldots z_n\}$ constructs an analytical function $z = f(x, y)$, which is a space surface. The function values of points other than known points can be calculated through these points. The specific significance of the point set in this study is as follows: $x_i$ represents the longitude of the center point of the grid, $y_i$ represents the latitude of the center point of the grid, $z_i$ is the ground change monitored by InSAR. The starting reference surface is 6 August 2021, and the ending reference surface is 9 May 2022. Through SAR data processing, the phase change is converted to the vertical change on the ground (the annual change rate is the total annual change) as a function value in the z direction. Its generalized regular grid and scattered node interpolation methods are shown in Figure 4II,III:

Rules for grid interpolation processing method, through the use of the space point $(x_1, y_1, z_1), (x_2, y_2, z_2), \ldots, (x_n, y_n, z_n)$, the scatter plot obtained by projecting on xOy plane is drawn equidistant along $x$ and $y$ axes. The resulting grid is a regular grid with $m \times n$ nodes $(x_i, y_i, z_{ij})(i = 1, 2, \ldots, m; j = 1, 2, \ldots n)$, where $x_i$ and $y_j$ are different. Suppose:

$$a = x_1 < x_2 < \ldots < x_m = b \tag{10}$$

$$c = y_1 < y_2 < \ldots < y_n = d \tag{11}$$

By constructing a binary function $z = f(x, y)$, by all the known node, that is, $f(x_i, y_j) = z_{ij} (i = 0, 1, \ldots, m; j = 0, 1, \ldots n)$ can be used to calculate the interpolation using $f(x, y)$, namely $z^* = f(x^*, y^*)$. In this paper, the buffer obtained through the analysis of the seawall road is rasterized and its center point is extracted as the data point of the regular grid.

Compared with the interpolation of regular grid, irregular grid search only requires that each node is not the same. The scattered nodes are irregular and the interpolation is difficult. In this paper, the fusion of PS point and SBAS treated grid center point is the irregular grid scattered node. The principle is as follows: Given n nodes $(x_i, y_i, z_i)(i = 1, 2, \ldots n)$ of which the $x_i\, y_i$ each other is not the same, by constructing a binary function $z = f(x, y)$, by all the known node, that is, $f(x_i, y_i) = z_i (i = 0, 1, \ldots, n)$, using $f(x, y)$ to calculate the interpolation, namely $z^* = f(x^*, y^*)$.

Given fusion TS-InSAR results point of discrete degree and quality, this article adopts the method of the adjacent interpolation and linear interpolation for the loss of coherent area seawall road interpolation fill the optimization, the adjacent interpolation method (Figure 4IV) according to the interpolation node to every other interpolation nodes, the interpolation node of the minimum of the function value is the required the interpolation results, The corresponding function value of the interpolation node closest to the interpolated point is the interpolation result. The process of piecewise linear interpolation (Figure 4V) is as follows: connect the diagonal lines of the square together, and divide the square into two triangles. Each square is treated this way, and can be viewed as a Mosaic of triangles. Each of the three points corresponds to the projection of three points in space. These three points are different; the three points in space are also different. In space, three points that are not collinear can uniquely determine a plane, and a space surface can be viewed as a Mosaic of multiple triangular space planes. Each plane corresponds to a binary linear function, so it is called fragment linear interpolation, Bilinear interpolation (Figure 4VI) is composed of quadric surfaces in space, and its function expression can be expressed as follows:

$$f(x, y) = (ax + b)(cy + d) \tag{12}$$

There are four undetermined coefficients in the expression, and four undetermined equations can be obtained by using the function at the four vertices (interpolation nodes) of the rectangle in Figure 4VI, so as to determine the four coefficients and obtain the function expression.

As the deformation data cannot be accurately obtained in the region of the soil embankment where there is no data for incoherence, this paper provides a referential fusion interpolation filling scheme, That is, four methods are used alternately, including the mean value of the sequence, the mean value of the adjacent points, the median value of the adjacent points, and linear interpolation. The existing sequence is regressed on an index variable scaled from 1 to n, and the missing value is replaced with its predicted value. The interpolation flow scheme is designed for areas with light and severe incoherence, as shown in Figure 5.

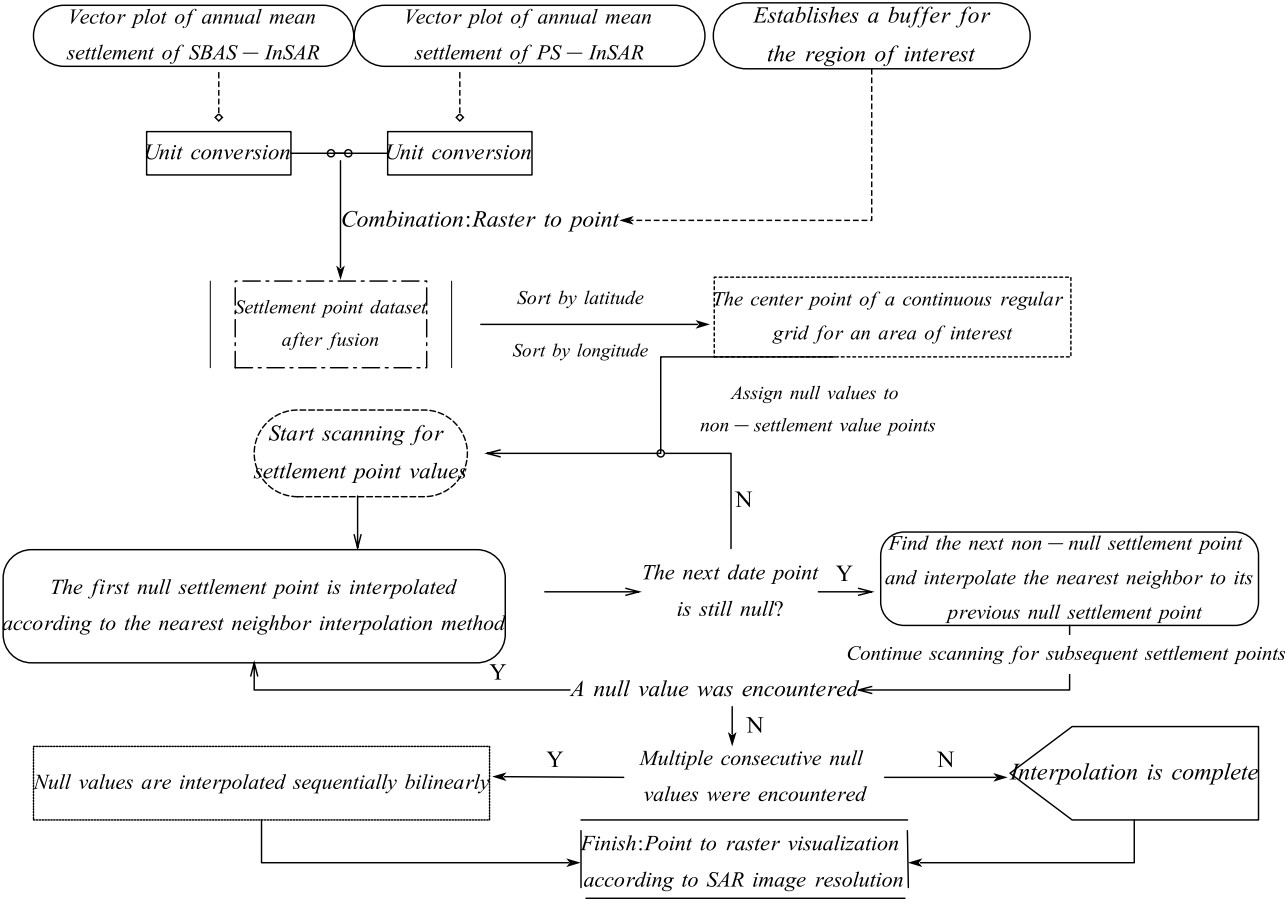

**Figure 5.** Interpolation flow scheme for missing data areas.

## 3. Results and Analyse

### 3.1. D-InSAR Spatial and Temporal Analysis of Sludge Area

In view of the analysis and evaluation of the sea sand reclamation work area in the area, a one-year monitoring was carried out. The selected time phase of the monitoring period was approximately consistent with the monitoring period of leveling survey, and D-InSAR monitoring was carried out simultaneously with it [39], Some parameters of the SAR image are shown in Table 2:

**Table 2.** Level measurement time and main parameters of SAR imagery.

| Level Measurement Time | Master Image | Slave Image | Imagery Parameters | Time Baseline/d | Spatial Baseline/m |
|---|---|---|---|---|---|
| 20210810 | 20210806 | 20211017 | | 72 | −19.580 |
| 20211017 | 20211017 | 20211204 | | 48 | 41.365 |
| 20211208 | 20211204 | 20211228 | | 24 | 26.748 |
| 20211226 | 20211228 | 20220121 | File type: SLC Polarization: VV Beam Mode: IW (TOPS mode) Direction: Ascending Subtype: SA Angle of incidence 21° | 24 | 12.218 |
| 20220122 | 20220121 | 20220202 | | 12 | 56.617 |
| 20220204 | 20220202 | 20220226 | | 24 | −139.093 |
| 20220225 | 20220226 | 20220310 | | 12 | 17.996 |
| 20220309 | 20220310 | 20220403 | | 24 | −55.028 |
| 20220403 | 20220403 | 20220509 | | 36 | 76.386 |

The SAR image data processing is carried out in two ways. The continuous settlement analysis is based on the SAR image of 2021.8.6 when the sea sand reclamation starts in the new sediment area around the Mud area. and the auxiliary image is 2021.10.17. The main image processed by D-InSAR for each time was used as the auxiliary image of the upper group, and the shape variables of each successive period were obtained by connecting them:

According to the D-InSAR continuous observation results in the Mud area, the area was continuously filled in every period from 2021.8.10 to 2022.5.9. From 2022.4.3 to 2022.5.9, the maximum filling thickness of sea sand was detected to be 10.02 cm. It is located near the cofferdam area in the northeast of the sludge area (dark red area in Figure 6i). During the period of 2022.2.2 to 2022.2.26 and 2022.2.26 to 3.10, the amount of sea sand accumulation near the seawall road in the study area increased significantly. In order to more intuitively explore its impact on the stability of the seawall road, Again, the SAR image with time phase 2021.8.6 is used as the main image. 2021.10.17, 2021.12.4, 2021.12.28, 2022.1.21, 2022.2.2, 2022.2.26, 2022.3.10, 2022.4.3 and 2022.5.9 were used as auxiliary images to detect the total ground height change of D-InSAR As can be seen from the cumulative sea sand filling accumulation diagram (Figure 7), during the whole period of sea sand and mud filling, uneven accumulation occurred.

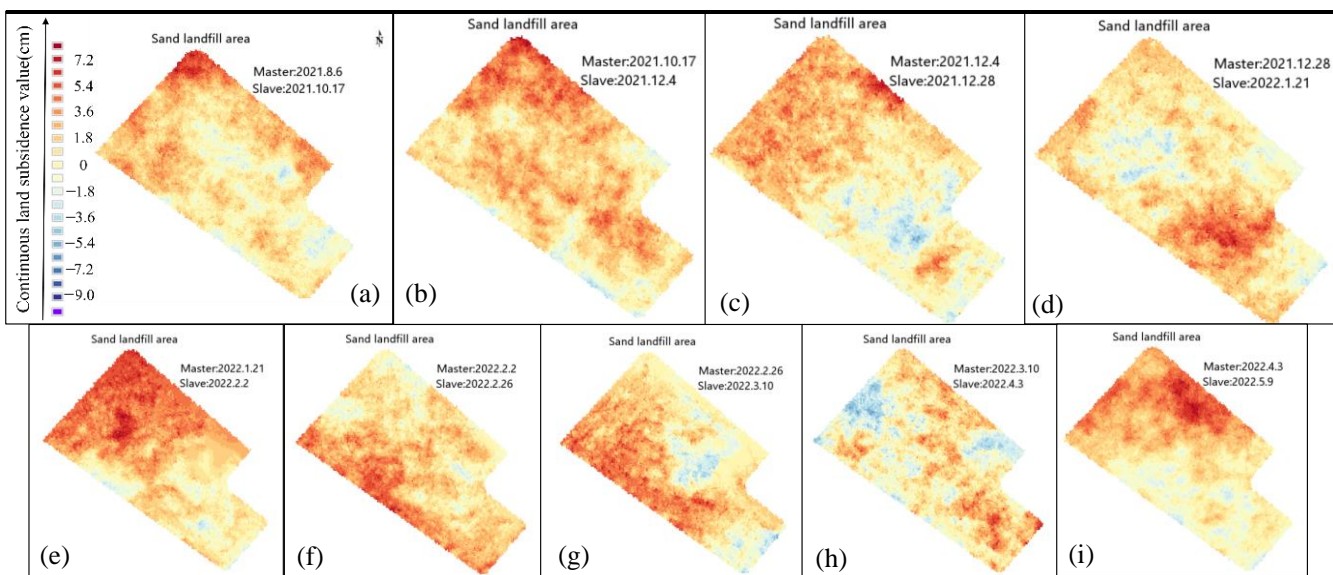

**Figure 6.** The amount of ground change (subsidence/cm) detected by D-InSAR continuously and periodically during the process of sea sand reclamation in the Mud receiving area. (**a**–**i**) plots correspond to different continuous time intervals.

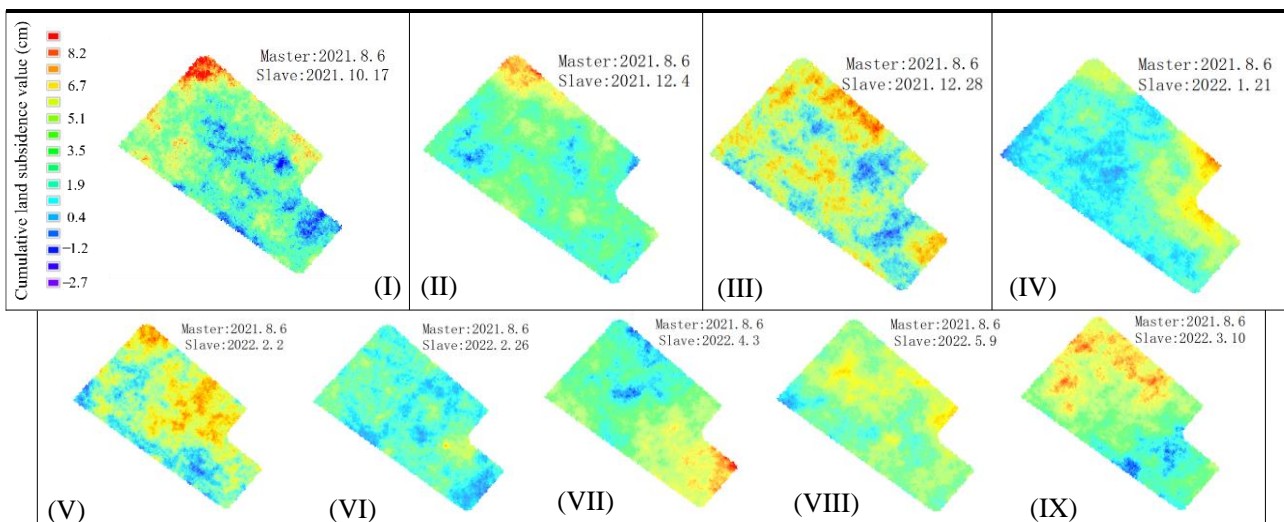

**Figure 7.** Cumulative subsidence of sea sand filling in the mud receiving area detected by D-InSAR/cm. (**I–IX**) plots correspond to time intervals for different cut-off times.

According to the changes of ground height in this area in different times, the deformation velocity field in the sludge area was made (Figure 8).

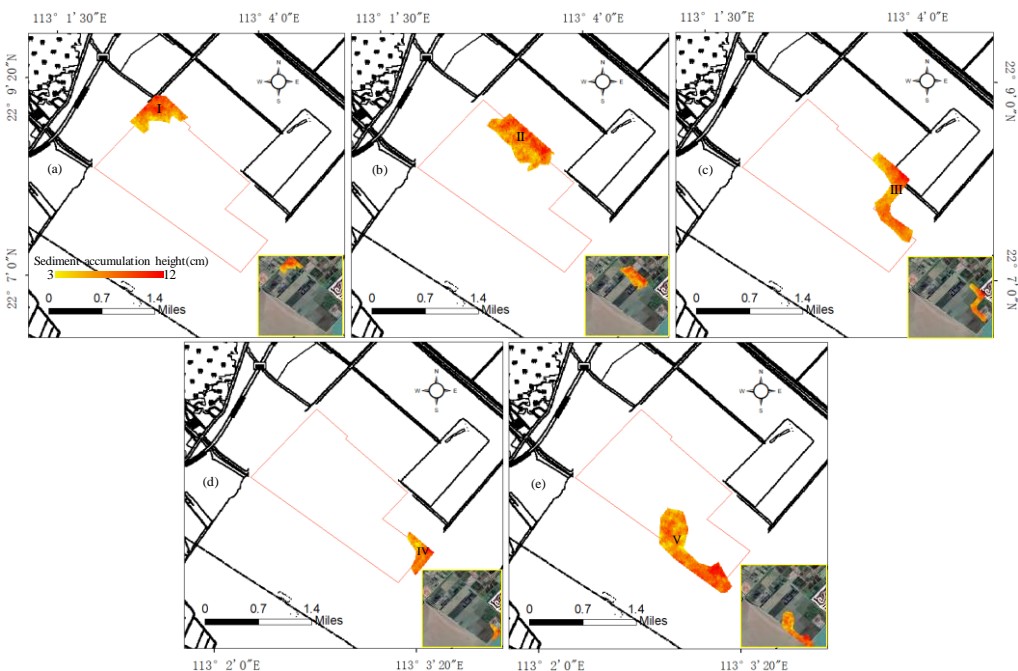

**Figure 8.** (**a–e**) plots show several areas of sediment accumulation observed by D-InSAR and sand flows in mud reception zones during monitoring.

From 2021.8.6 to 2022.3.10, we observed five important sediment high density accumulation areas, namely I, II, III, IV and V, respectively. As shown in Figure 8, the sediment accumulation area was always in a state of sediment discharge during the whole cycle. With the filling of sediment, the sediment accumulation area spread and accumulated alternately. The maximum height has exceeded 10 cm, as shown in Figure 9, among which I and V accumulation areas are around the road, which brings pressure to the stability of the road:

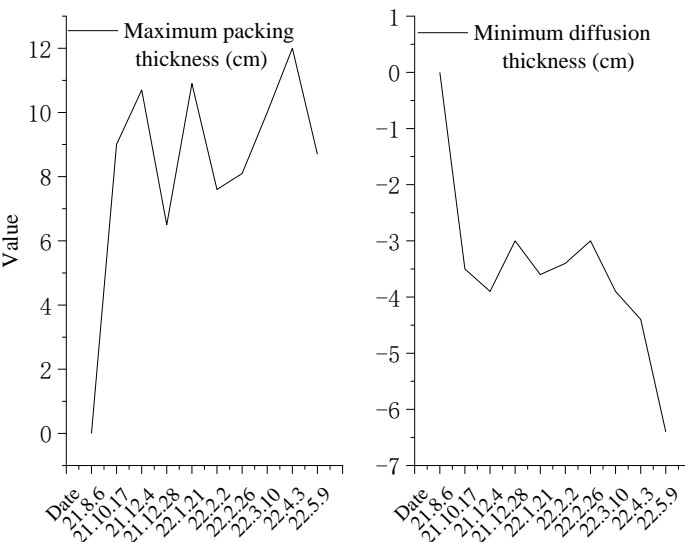

**Figure 9.** Cumulative maximum and minimum changes in ground height in the Mud receiving area observed during D-InSAR observations.

Influenced by the temporal and spatial baselines, the SAR image time baseline of the Mud area from 6 August 2021 to 9 May 2022 was estimated to be 276 days through the baselines. According to the data provided by the Central Meteorological Observatory, the rainfall concentration period in the sludge area was from May to September, accounting for more than 75% of the whole year. The average number of days with precipitation accounts for about 40% of the whole year. The variation of soil water content caused by precipitation in the sludge area, as well as the irregular area and thickness of sea sand blowing and filling during this period, lead to great surface deformation. These factors make the physical and chemical properties of objects in the observation area resolution unit randomly change with time during the two radar imaging periods, resulting in temporal and spatial incoherence (Figure 10).

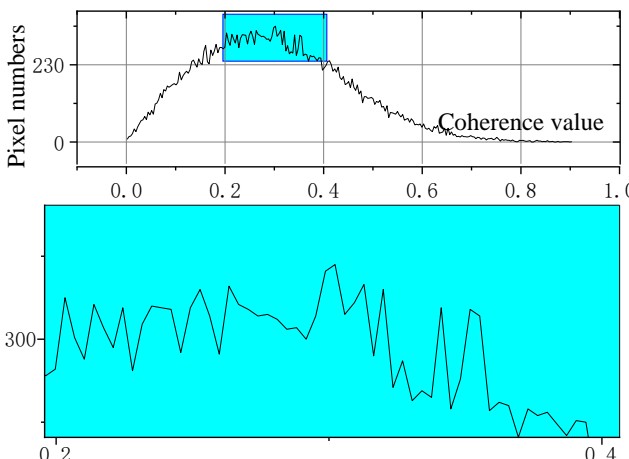

**Figure 10.** 2021.8.6 to 2022.4.3 SAR image coherence value and corresponding number of pixels.

For comparative accuracy analysis, Through precise coordinate transformation of the coordinate value of monitoring points [40,41], gray values of pixels at corresponding positions of 8 feature points, such as H1, H3, H5, H8, H11, H14, H17 and H21, in part of Figure 2, are extracted. Given that the pixel resolution of SAR image is about 15 m, the coordinate conversion error is decimeter level. Therefore, the pixel pick point position error of feature points is negligible, and it is compared with the change value of continuous observation and cumulative observation of third level respectively.

According to the comparative data in Figure 11, during the continuous observation period, the maximum difference between the settlement amount of point H21 observed by D-InSAR and that observed by horizontal survey from 21 January 2022 to 2 February 2022 is 36.9 mm, far exceeding the warning value of ±10 mm of seawall road stipulated in Table 1. In the cumulative observation and settlement process, From 6 August 2021 to 3 April 2022, the difference between the accumulated settlement value of H3 point monitored by D-InSAR and the leveling result is 78.5 mm. According to the coherence value in Figure 10, the coherence of SAR image is generally concentrated at 0.2 in this period, and the image incoherence is serious. It interferes with part of the results detected by D-InSAR and affects the stability evaluation of seawall road, cofferdam and earth embankment [42]. Therefore, in order to detect more precisely the influence of sea sand casting on the stability of seawall road, high-precision Time Series InSAR (TS-InSAR) processing is adopted.

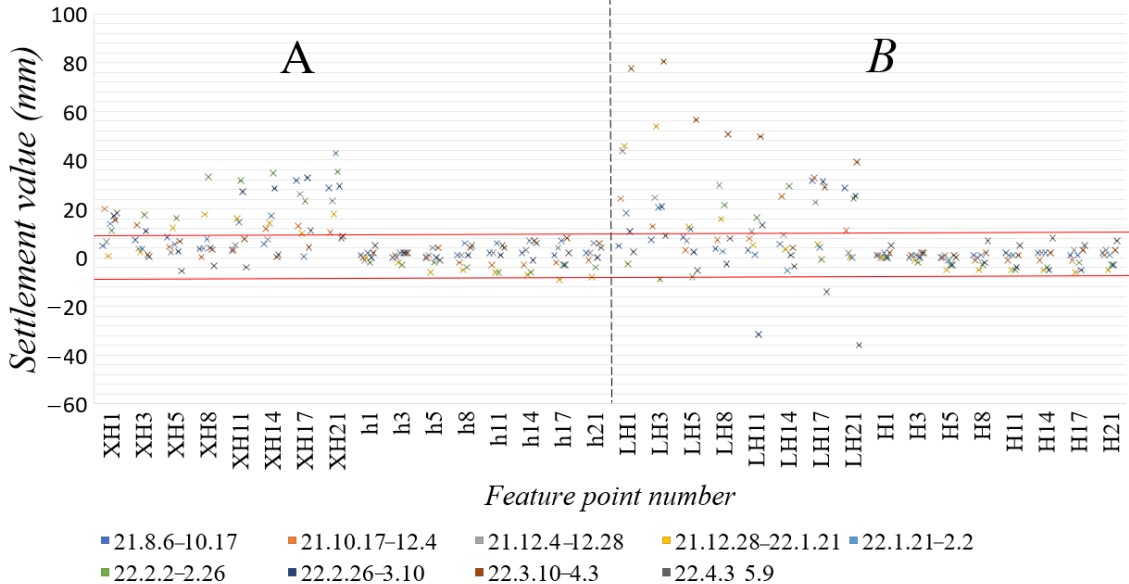

**Figure 11.** Comparison of the continuous subsidence (**A**) and cumulative subsidence (**B**) measured by some leveling monitoring points and the corresponding subsidence of D-InSAR (H1-H21 are the leveling monitoring points, XL1-XL21 are the corresponding D-InSAR detections The position of the image element point of the continuous deformation result, LH1-LH21 is the corresponding image element point position of the cumulative deformation result detected by D-InSAR, the range between the two red lines is the reasonable settlement interval, and the outside the red line is the vigilant settlement value).

### 3.2. Inversion of Seawall Road Settlement Using TS-InSAR

The time phase of the SAR image wasselected from 24 continuous SAR images from 6 August 2021 to 9 May 2022, before the start of the sludge area filling. By connecting multi-temporal SAR data, several interference image pairs were formed. The connection diagram is shown in Figure 12, and the common main image time phase is 14 February 2022.

Estimation accuracy of 4.82168 mm/year, height accuracy of 8.37999 m, Each coherence interference image threshold is set as 0.8, remove the correlation coefficient is less than the threshold value of the pixel, its not preserved in PS results, get 223 focused on PS permanent scattering point for a mud area seawall road (Figure 13a yellow area), the average annual settlement on +/−5 mm/a, through PS permanent scattering point the number and distribution of the image, Its coherence higher concentrations launched few PS points and road sections near southeast, through on-the-spot investigation understands, mud area surrounding the earth embankment vegetation lush, the echo signal is poorer, cofferdam accumulation area for soft soil area, the monitoring period is always covered with black waterproof fabric, time and increase the amount of soil is solid, coherent severe

loss in the area image, Seawall road surrounding concrete protective casting (Figure 2) the radar backscatter is stronger, for mud receiving area for land reclamation phase fluctuation on the ground, stability region of the mainland on the sequence of volatile, causing loss of coherent imaging, the land surrounding the earth embankment and cofferdam for lack of high echo signal feature, then the scattering intensity is weak, The PS point cannot be detected.

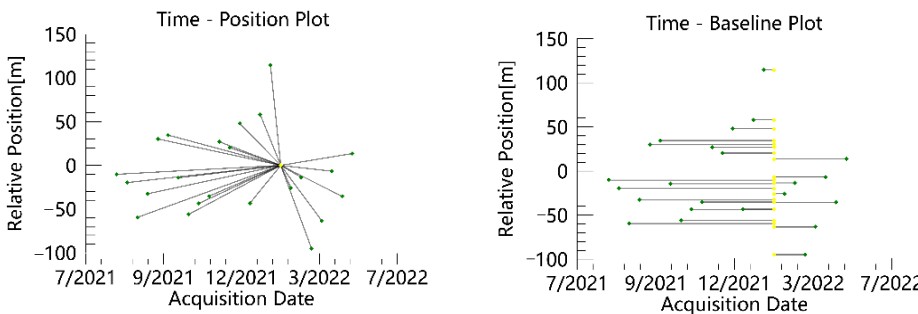

**Figure 12.** Time Baseline and Common Main Image Selection (green—valid data pairs; yellow—super main image).

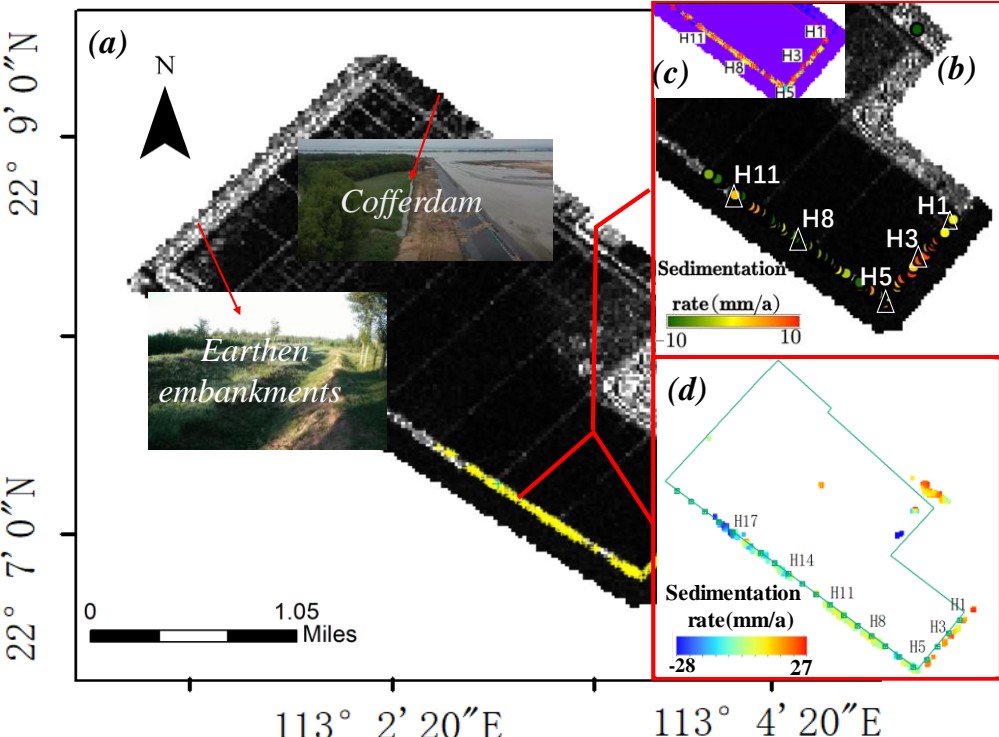

**Figure 13.** TS-InSAR deformation velocity field in the sludge area (The yellow part of (**a**) is the PS point in the sludge area, (**b**) is the deformation rate of PS point in the vertical direction (unit: mm/a),H1-H11 is the leveling monitoring point, (**c**) is the average deformation situation in this area and the relative position of the leveling point. (**d**) is the vertical deformation of the ground obtained by SBAS-InSAR).

According to the average radar intensity map of the subsidence velocity field of the sea wall road in the mud-bearing area, the surface subsidence in the study area is relatively uniform in spatial distribution, which can be reflected by the almost uniform color in Figure 13c. The settlement in cofferdam area is significant, with the maximum settlement reaching −16 mm/a, but not exceeding the warning standard value defined in Table 1. The overall settlement around the seawall road is relatively small, with an average of about −2 mm/a. The PS points cover the level monitoring points H1, H3, H5, H8 and H11. In

PS-InSAR, the nearest point to the level is taken as the comparison point of the level, among which PS1, PS3, PS5, PS8 and PS11 are the extracted settlement feature points uniformly distributed near the sea wall road, and these feature points are the PS points closest to the level monitoring point. These PS points and SBAS feature points are used to analyze the time series of seawall road sections with high backscattering intensity.

During the period from 6 August 2021 to 9 May 2022, the settlement near the char acteristic point H1 section of seawall road fluctuated within the range of ±5 mm in the whole process of sea sand filling, as shown in Figure 14. The settlement of the section of H3 monitoring point was 11.68 mm on 5 October 2021, exceeding the warning value of 1.68 mm, and fluctuated within the normal range in the subsequent time, which could be used as one of the later route maintenance sections. The section of H5 monitoring point was in the overall uplift process during the monitoring period. Affected by the extrusion of Area IV, an important sediment accumulation area formed on 3 April 2022 in the sludge bearing area (Figure 9), this section experienced a relatively large settlement in the following time. The maximum settlement reached 9.28 mm on 3 April, not exceeding the warning value, and the overall fluctuation of section H8 in the monitoring point was within the range of ±5 mm.

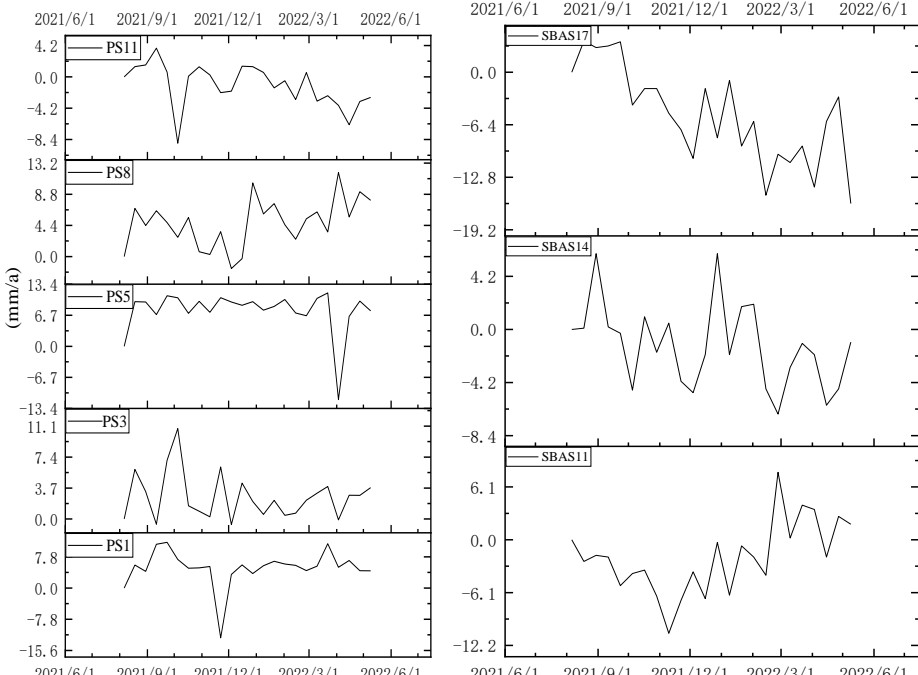

**Figure 14.** Overall settlement law of PS points near the embankment (PS11, PS8, PS5, PS3, PS1, SBAS17, SBAS14, SBAS11 corresponding to points H11, H8, H5, H3, H1, H17, H14, H11respectively).

For the purpose of precision analysis, it was compared with the settlement value monitored by the level, as shown in Figure 15. The maximum difference between PS-InSAR monitoring results and leveling monitoring results is 1.59 mm/a, and the minimum difference is 0.71 mm/a. Combined with the error of ±2 mm in third-class leveling measurement, it can be seen that PS-InSAR measurement results reach the safe settlement warning value of ±10 mm within the maximum allowable error range. PS-InSAR analysis shows that the cofferdam and earth embankment in the sludge area are seriously incoherent and cannot be monitored according to PS-InSAR. SBAS air filter in the inversion process coherence threshold is 0.2, the existing of SAR image data set to form a number of small collection, increase the amount of data the H10-H17 section, the The average settlement amount of seawall road in this period is shown in Figure 16, and the time series combined with leveling results and PS-InSAR comparative analysis in the monitoring period is shown in Figure 14. Through the settlement rate comparison of public monitoring points, The

maximum difference generated by the three detection methods is 1.77 mm/a, which is the difference between SBAS H18 point and leveling measurement. The settlement of this monitoring point obtained by SBAS method is 6.77 mm/a. It is considered that the settlement of HaiDi road monitored by SBAS can meet the requirements of accuracy within the allowable range of error.

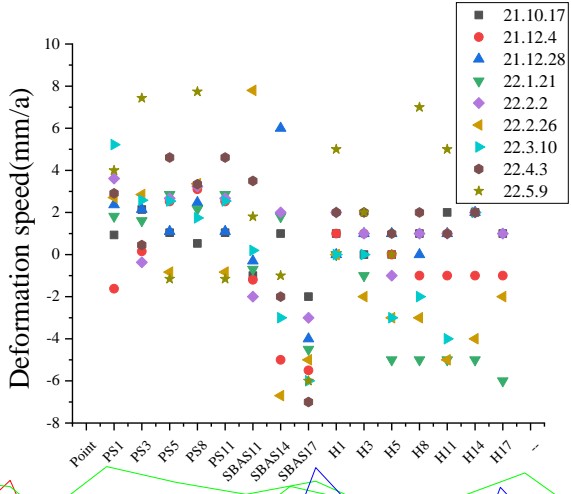

**Figure 15.** Relative accuracy of TS-InSAR measurement results and leveling results.

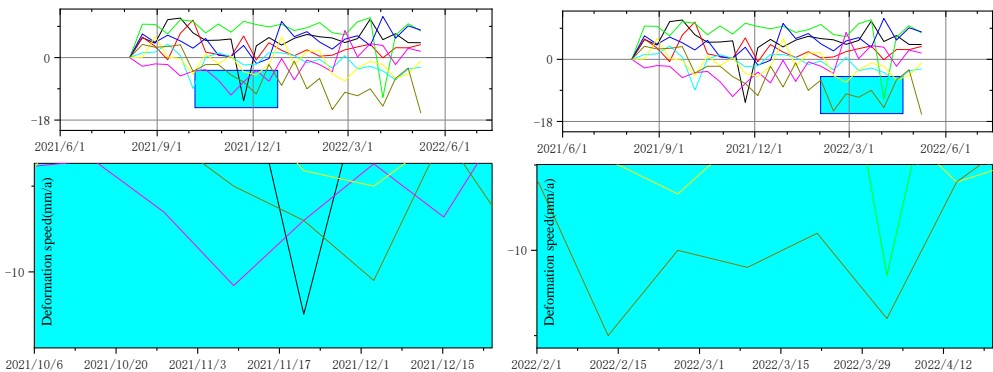

**Figure 16.** Time sequence analysis of SBAS10-SBAS17 of the monitoring feature points of the southwest section of Haidi Road in the mud area.

By observing the time series diagram, it can be seen that the settlement rate of the H11 level monitoring point in the silt absorption area in the whole period is about 2.09 mm/a on the weighted average of the three methods. The settlement rate of the H11 level monitoring point in the silt absorption area is affected by the large area of silt filling accumulation in the silt absorption area (Figure 7III) on 10 November 2021, and the maximum settlement warning peak of this section is −10 mm/a. Subsequently, due to the diffusion of sediment, this section of road steadily rises to the normal rate, and the section of monitoring point H14 is in the process of steady uplift and settlement during the monitoring period. The average settlement rate is about −0.82 mm/a. The section of monitoring point H17 has a large settlement due to the influence of massive sediment accumulation and extrusion on this section from 2 February 2022 to 10 March 2022 (Figure 6f,g), which is about 6 mm beyond the settlement warning value, and continues to exceed the limit range in the subsequent time. The settlement rate is −15.36 mm/a at the end of the monitoring period, so it is necessary to maintain and reinforce this section to ensure the safety of the road.

### 3.3. Integration Feasibility Assessment

The sedimentation rate was taken as the dependent variable, and the public monitoring points were selected as H1, H3, H5, H8, and H11. The measurement method and monitoring point number were used as factors for normality statistics, this is shown in Figure 17:

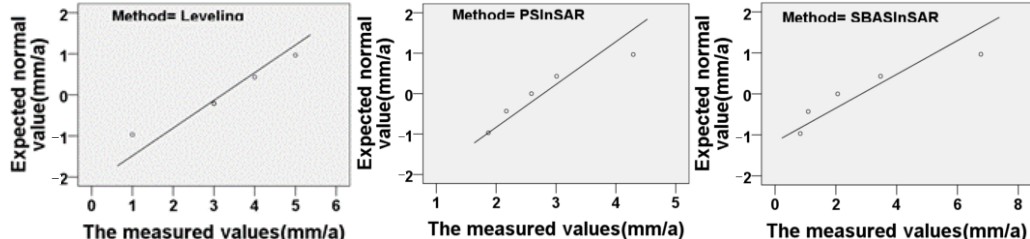

**Figure 17.** The difference between the three measurements and the normal expected value.

Through the Shapiro-Wilk significance of 0.777, 0.524 and 0.26 is far greater than 0.05, it can be found that there is normality between the result of leveling measurement and the result of interferometry for the sedimentation rate of the dependent variable. It can be considered that the deviation between the measured value and the normal value of each method fluctuates in a small range. The variance analysis of the sedimentation rate is shown in Table 3.

**Table 3.** Inter-subject effect test.

| Dependent Variable: Settlement | | | | | |
|---|---|---|---|---|---|
| The Source | Class III Sum of Squares | Degrees of Freedom | The Mean Square | F | Significant |
| Correction model | 28.722 | 6 | 4.787 | 4.972 | 0.021 |
| intercept | 129.948 | 1 | 129.948 | 134.969 | 0.000 |
| Point | 28.219 | 4 | 7.055 | 7.327 | 0.009 |
| Method | 0.502 | 2 | 0.251 | 0.261 | 0.777 |
| error | 7.702 | 8 | 0.963 | | |
| total | 166.373 | 15 | | | |
| Total after correction | 36.424 | 14 | | | |

According to statistics, the mean values of the three monitoring methods for the monitor ing points are different, which are 3.2 mm/a, 2.79 mm/a and 2.84 mm/a respectively. It can be considered that the monitoring effects of different monitoring methods are slightly different, and the measurement results have high credibility and good consistency. According to the significance of detection point and measurement method 0.009 and 0.777, since the monitoring point is the same public point, it is considered that there is no significant difference between them [43,44]. Since the significance between measurement methods is greater than 0.05, the three measurement methods cannot be considered different, which further improves the reliability and matching degree of monitoring results.

### 3.4. Optimization and Interpolation

Through the results of variance analysis, we verified the matching of the results of PS-InSAR and SBAS-InSAR. In order to fully analyze the influence of sand filling on pavement and maintenance embankment in mud-bearing area, grid data of processing results were extracted from the grid center point to obtain the coordinates and deformation information (X, Y, Vertical deformation rate) of each settlement point. We integrated the PS points processed by PS-InSAR and the grid center point data obtained by SBAS-InSAR, and generated a new pavement settlement according to the image resolution of Setinel-1A, as shown in Figure 18b.

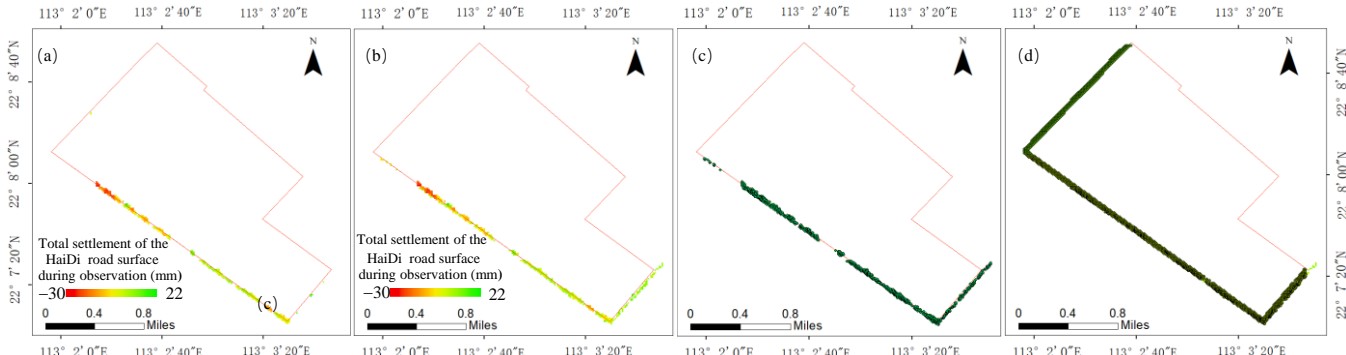

**Figure 18.** (**a**) the road settlement monitored by SBAS-InSAR (**b**) The road settlement obtained after the fusion of PS-InSAR (**c**) the grid center points extracted by fusion InSAR (**d**) the buffer established according to the road width and the grid center points extracted.

By fusion of TS-InSAR deformation information, preliminary is improved after the pavement deformation of the image, to facilitate the construction and application of interpolation model, we of the seawall road and soil embankment pavement buffer is analyzed, based on its pavement width buffer is established, and its resolution SAR resolution remain the same, to extract the center of each buffer grid, As shown in Figure 18d, the missing of the entire pavement is processed according to the process in Figure 19.

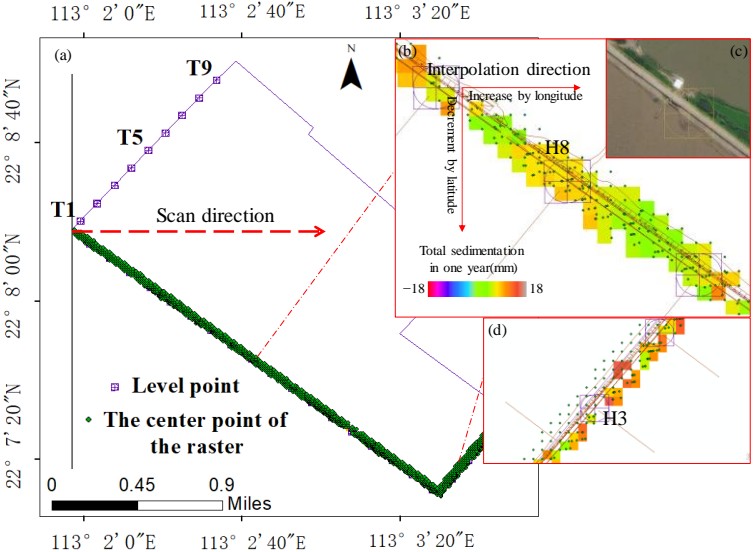

**Figure 19.** (**a**) All grid centers with settlement values and the overall interpolation direction of the seawall road section. T1-T9 is the 9 level monitoring points on the soil embankment road. (**b**) Interpolation examples near the level point H8 of the Seawall road are the real settlement area after InSAR fusion. The red line is the line map of the road terrain (**c**) the optical image of the road near H8, and (**d**) the settlement law of the H3 level section of HaiDi Road, whose default value type is completely missing.

In order to verify the accuracy of adjacent interpolation and linear interpolation, the total deformation of 21 level monitoring points H1-H21 on Seawall Road and 9 level monitoring points T1-T9 on Earth Embankment Road from 10 August 2021 to 6 May 2022 is used as a reference. In order to solve the not only tonality in longitude direction of Earth embankment road and sea Embankment road, Of earth dam and seawalls road we break up, as shown in Figure 19a, figure in the green dot represents the fusion of InSAR settlement point, two sections of split ensures the interpolation direction, continuity and monotonicity, these points through value sorted according to the longitude and latitude,

formed a fixed scanning sequence, as shown in Figure 19b,c the graph is the image of the area, the interpolation order according to the settlement of monotonous sequences in a row, according to the longitude increase, according to the latitude decreases, and formation rules of the interpolation direction of continuous monotonous, guarantees the domain of continuity, also ensure the pavement can be handling [45], among them the main lack of road surface subsidence data types as shown in Figure 19b,d shown below: The deletion types are adjacent deletion and complete deletion.

Aimed at the default value of the two main types we through the process of algorithm in the design of Figure 5 for interpolation processing, as shown in Figure 20.

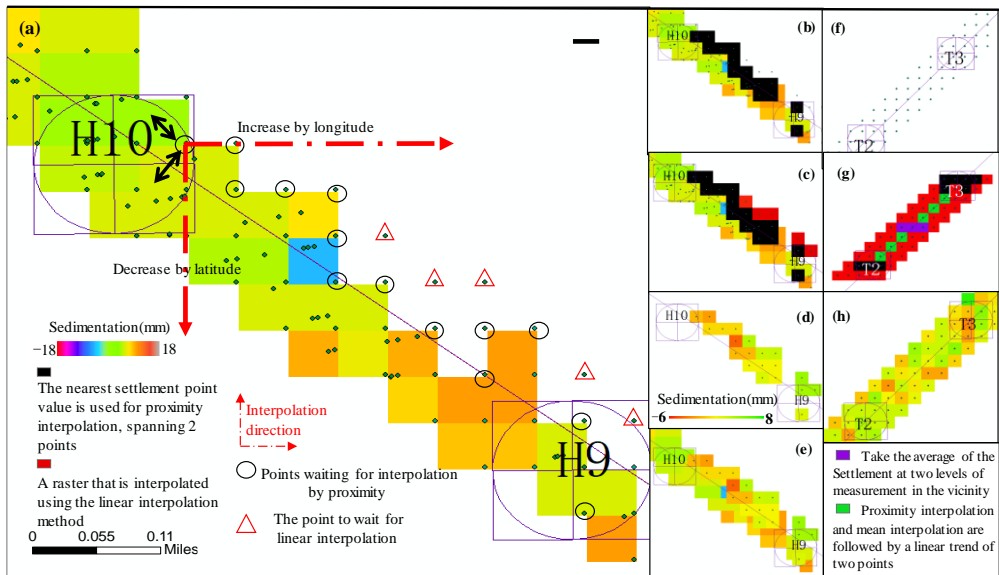

**Figure 20.** (**a**) the deformation of H9-H10 section of Seawall Road pavement, (**b**–**e**) the first type of adjacent edge missing, interpolation process and results using the adjacent method, (**f**–**h**) the type of completely missing road deformation data of the second type and its processing method.

For the first kind of missing value according to the most adjacent interpolation and bilinear interpolation method of combining the, as shown in Figure 20a as shown in figure, black circle part adopts the adjacent interpolation method of interpolation points interpolation, to around the null value as a reference for interpolation, Due to the non-null settlement points around the red triangle, the value of the adjacent interpolation points is used for bilinear interpolation. The specific interpolation sequence is shown in Figure 20b,c. The black square is the ground area for interpolation using the adjacent interpolation method, and the red square is the ground area for interpolation using the bilinear interpolation algorithm. After the completion of interpolation, the real deformation data fused with InSAR were used for unified visualization, including visualization of resolution and color hue, to achieve the interpolation encryption target of missing adjacent edges. Figure 20e shows the encrypted deformation data of the section after encryption.

In order to facilitate the analysis of the overall settlement of the soil embankment road and seawall road section, also, simple interpolation between T1–T9 level points of the embankment road section is carried out, as shown in Figure 20f–h.

## 4. Discussion

In order to verify the settlement interpolation accuracy of the road block area, we verified the data of all level monitoring points of seawall road and soil embankment road, and extracted the deformation values of the roads in the areas where these level monitoring points were located for comparison, as shown in Figure 21.

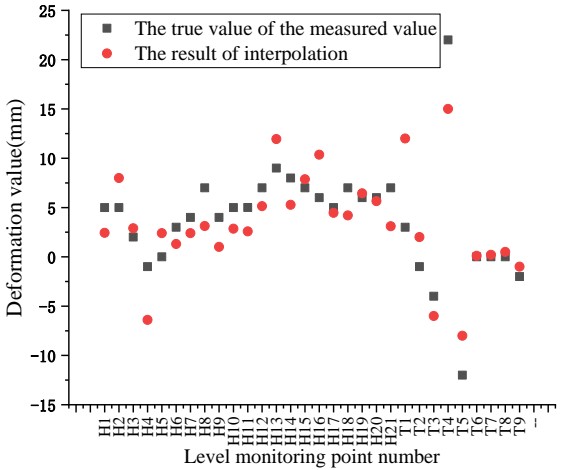 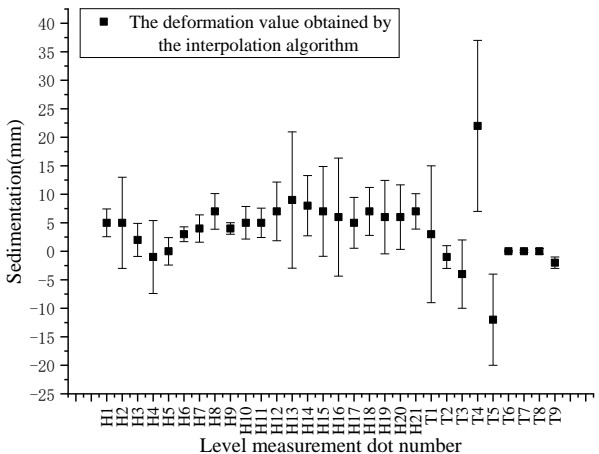

**Figure 21.** Comparison of real values and interpolation results of level monitoring points on seawall road and soil embankment road (Figure right) error.

By comparing the measured deformation value of the leveling point with the deformation value of the interpolation area, we found that the maximum difference between the two was 7 mm and the minimum difference was 1 mm. the total deformation of the level monitoring point on the overall road was in good agreement with the grid deformation value corresponding to this point. Within the allowed error range of satellite image resolution [46,47], it can be considered that the two interpolation methods maintain the continuity of road surface, and the surrounding deformation obtained by these leveling monitoring points as interpolation basis points can initially replace the continuous deformation of road surface. In order to facilitate the analysis of the settlement deformation of the whole section, According to the settlement warning value and deformation level shown in Table 1, a settlement map was made for the total deformation of seawall road and earthen embankment road from 10 August 2021 to 9 May 2022, as shown in Figure 22.

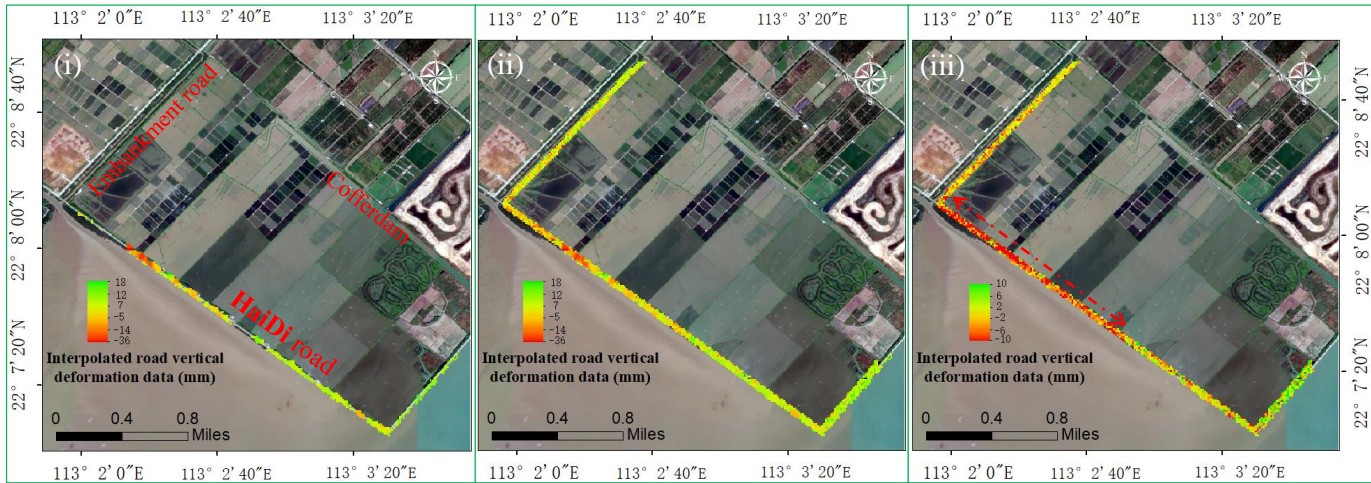

**Figure 22.** (**i**) Pavement deformation and settlement map integrating PS-InSAR and SBAS-InSAR (**ii**) Pavement deformation corrected by proximity interpolation and bilinear interpolation (**iii**) the image with deformation and settlement range adjusted to −10–10 (mm).

The overall deformation of the two sections in the mud-containing area after interpolation optimization is shown in Figure 22iii. Combined with the settlement threshold in Table 1, the deformation and settlement data are adjusted to −10–10 mm. It can be found that the northwest section of the Seawall Road is in a slight settlement level as a whole,

and the rest sections of the soil embankment road and the Seawall road do not have large settlement and uplift phenomenon. To facilitate the understanding of the specific settlement level of the section, the initial pavement height before the project starts on 6 August 2021 is taken as the initial 0 reference height, and the ground deformation on 9 May 2022 is taken as the change height to draw the three-dimensional DEM and DEM profile of this area, as shown in Figure 23.

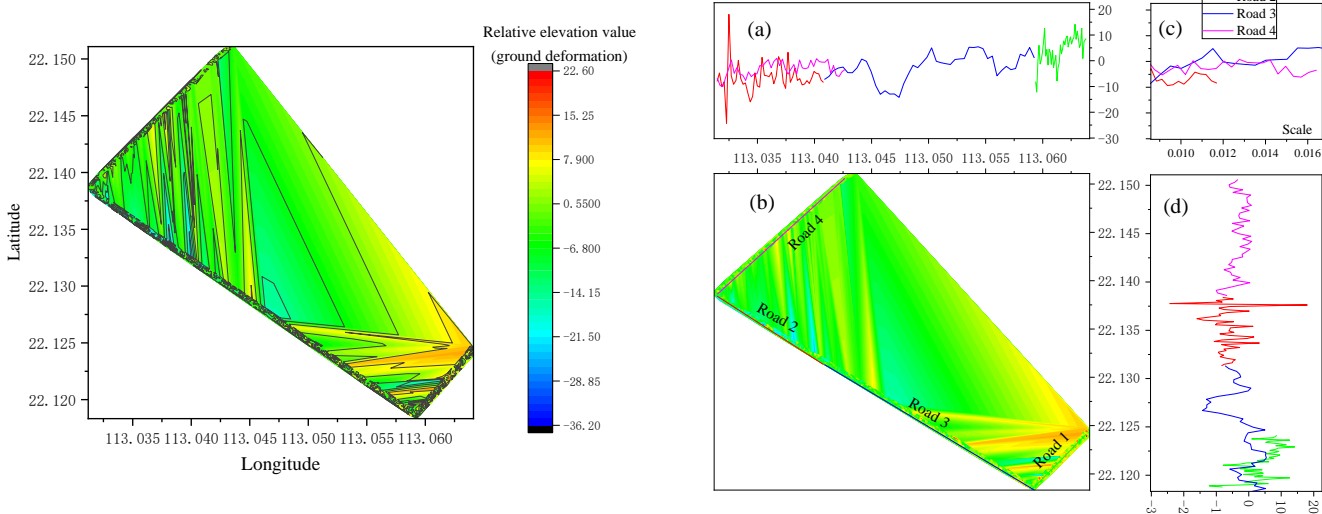

**Figure 23.** DEM and DEM profile of the surrounding roads in the Mud receiving area after interpolation optimization. (**a**,**c**) show the profile of the DEM in both latitude and longitude, (**b**) shows the DEM of the area, and (**d**) shows the scales of different roads.

Figure 23 left shows the maximum settlement and lift height of seawall road and soil embankment road in the whole monitoring cycle of 36.2 mm and 22.6 mm; Figure 23 right shows the DEM profile of this section. According to the settlement law of the road surface in Figure 22iii, the road surface is divided into four sections, namely road 1, road 2, road 3 and road 4, as shown in Figure 23b. Figure 23a,c are two projectors respectively. Figure 23d shows the length and position of the four roads. According to the deformation of the roads on two projectors Figure 23a,d, the roads around the mud area are in a slight settlement level as a whole, some roads have uneven settlement, and the settlement deformation of road 2 fluctuates greatly. Within the whole cycle, the maximum settlement of this section is about 25 mm, and the maximum uplift is more than 15 mm. It is located at 113.031° E–113.037° E, 22.135° N–22.14° N, and the settlement level is relatively serious. Therefore, the road warning and maintenance design should be carried out in advance, and the verification should be combined with the field survey results [48]. If not paid attention to, it is likely to affect the comfort of driving and durability of the road surface.

## 5. Conclusions

Human irregularities bring the risk of natural disasters, as does the dredging of waterways, and only by fully handling the relationship between risk and land planning [49] can losses be minimized. In this paper, the deformation of the mud receiving area and surrounding roads is inverted by InSAR technology, and the unreasonable blowing and filling of sediment on the seabed causes the uneven accumulation of sand storage area, which affects the use of the sand storage area, and at the same time causes extrusion of the surrounding traffic roads, causing uneven settlement of the road, and the settlement rate of some sections is too large and needs maintenance. On the basis of comprehensive time series InSAR, the settlement data is first fused and then interpolated, the density of settlement data is improved, compared with the level measurement data, the maximum

error of interpolation is about 7 mm, which is of great significance for road safety analysis within the allowable range of error, according to the results obtained by our processing in this paper, the following suggestions are put forward:

1. The dredging operation process of the channel should be reasonably standardized to avoid the occurrence of uneven accumulation, and the increase in sediment mobility will lead to the degradation of the surrounding land and the flooding of flooding.
2. Areas close to water are favorable locations for urban development, and the risk of mud areas increases the possibility of flooding, thereby causing disasters to surrounding fields and villages [50], and waterproofing projects such as rivers can be built according to demand.
3. Disaster assessment should be done in a timely manner for road sections with too fast road subsidence [51], such as coastal disaster vulnerability and social vulnerability index.

**Author Contributions:** Z.Q. conceived the manuscript; B.H. provided funding support and helped improve the manuscript. All authors have read and agreed to the published version of the manuscript.

**Funding:** Research on the key path of BIM technology embedded in the smart site management platform of the Guangdong Provincial Education Department in 2020 (Grant No. 2020KTSCX289). This work was supported by the "Research on the key path of BIM technology embedded in the smart site management platform" of the Guangdong Provincial Education Department in 2020 (Grant No. 2020KTSCX289).

**Data Availability Statement:** Sentinel-1A data used in this study were provided by European Space Agency (ESA) through the Sentinel-1 Scientific Data Hub. AW3D30DEM is freely provided by JAXA.

**Acknowledgments:** We are very grateful for the Sentinel-1A data provided by European Space Agency (ESA), Thanks for the support of CCCC South China Survey&Mapping Technology Co., LTD.

**Conflicts of Interest:** The authors declare no conflict of interest.

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
