# Peer review of "Spatiotemporal Characteristics of the Mud Receiving Area Were Retrieved by InSAR and Interpolation"

_remotesensing, doi:10.3390/rs15020351_

Round 1

Reviewer 1 Report

Revealing spatiotemporal characteristics of the mud receiving area is useful for monitoring the safety around the seawall road. Overall, the paper is well-written and easy to follow. There are some places in the paper need clarification or justification.

(i)              The term ‘D-InSAR technology’ first appeared in the abstract, explaining what is Dinsar would make the paper more readable.

(ii)            Does the red dotted line in Figure 1 represent Mud receiving area? Please make it clear. Moreover, adding legend into Figure 1 (not only the small part) could be useful.

(iii)          “If the deformation in the direction of SAR observation skew distance in the two SAR image information is set as” What is skew distance?

(iv)           “Figure left shows the maximum settlement and lift height of seawall road and soil embankment road in the whole monitoring cycle of 36.2mm and 22.6mm; Figure right shows the DEM profile of this section.” Which figure?

(v)             The flow chart of the paper (Figure 3) is complicated, perhaps a more concise flow chart is needed.

Reviewer 2 Report

The manuscript entitled Spatiotemporal characteristics of the mud receiving area were retrieved by InSAR and interpolation, by B. Hu & Z. Qiao, presents an interesting work.

In general, the manuscript should be acceptable for publication, but some serious problems must be repaired prior to publication. It needs some significant improvement. Some suggestions are as follows:

  1. Please use different terms in the “Title” and the “Keywords”.
  2. I suggest to shorten the conclusions section without sub titles.
  3. It would be useful to be described the aim of this paper.
  4. The English language usage should be checked by a fluent English speaker. It is suggested to the authors to take the assistance of someone with English as mother tongue.
  5. You could enrich the scientific literature.
  6. The authors could make discussion about the relationship between hazards and planning. See the following publication: “Land Use Planning for Natural Hazards”, Land, 2019, 8 (9): 128
  7. Correct references in the reference list according to the journal’s format. Please format the references’ list by using the correct journal abbreviations. See the following link: https://images.webofknowledge.com/images/help/WOS/A_abrvjt.html
  8. In the manuscript please use continually line numbering.
  9. Please be careful with the spaces between the words.

Reviewer 3 Report

Suggestion: "Spatiotemporal characteristics of the mud receiving area were retrieved by InSAR and interpolation".

The article is very interesting. My only remark relates to the quality of the figures (e.g. Fig. 5, Fig. 10, Fig. 11, Fig. 15, Fig. 21). They are properly described but illegible. Maybe increasing the resolution or changing the font size will improve their quality. The figures are placed once on the right and once on the left. Is that intentional (e.g. Fig. 11, Fig. 16, Fig. 21)?

Round 2

Reviewer 2 Report

The manuscript entitled “Spatiotemporal characteristics of the mud receiving area were retrieved by InSAR and interpolation”, by B. Hu, & Z. Qiao, presents an improved and good work.

The manuscript should be acceptable for publication in the present form.